# Reprogramming of Fundamental miRNA and Gene Expression during the Barley-*Piriformospora indica* Interaction

**DOI:** 10.3390/jof9010024

**Published:** 2022-12-23

**Authors:** Liang Li, Nannan Guo, Yanze Zhang, Zhi Yuan, Aidang Lu, Si Li, Ziwen Wang

**Affiliations:** 1School of Chemical Engineering and Technology, Hebei University of Technology, Tianjin 300401, China; 2Tianjin Key Laboratory of Structure and Performance for Functional Molecules, College of Chemistry, Tianjin Normal University, Tianjin 300387, China

**Keywords:** *Piriformospora indica*, miRNA, barley, RNA-seq, reprogramming, transcription factor

## Abstract

The interactions between plants and microorganisms, which are widely present in the microbial-dominated rhizosphere, have been studied. This association is highly beneficial to the organisms involved, as plants benefit soil microorganisms by providing them with metabolites, while microorganisms promote plant growth and development by promoting nutrient uptake and/or protecting the plant from biotic and abiotic stresses. *Piriformospora indica*, an endophytic fungus of Sebacinales, colonizes the roots of a wide range of host plants and establishes various benefits for the plants. In this work, an interaction between barley and the *P. indica* was established to elucidate microRNA (miRNA)-based regulatory changes in miRNA profiles and gene expression that occurred during the symbiosis. Growth promotion and vigorous root development were confirmed in barley colonized by *P. indica*. The genome-wide expression profile analysis of miRNAs in barley root showed that 7,798,928, 6,418,039 and 7,136,192 clean reads were obtained from the libraries of mock, 3 dai and 7 dai roots, respectively. Sequencing of the barley genome yielded in 81 novel miRNA and 450 differently expressed genes (DEGs). Additionally, 11, 24, 6 differentially expressed microRNAs (DEMs) in barley were found in the three comparison groups, including 3 dai vs. mock, 7 dai vs. mock and 7 dai vs. 3 dai, respectively. The predicted target genes of these miRNAs are mainly involved in transcription, cell division, auxin signal perception and transduction, photosynthesis and hormone stimulus. Transcriptome analysis of *P. indica* identified 667 and 594 differentially expressed genes (DEG) at 3 dai and 7 dai. Annotation and GO (Gene Ontology) analysis indicated that the DEGs with the greatest changes were concentrated in oxidoreductase activity, ion transmembrane transporter activity. It implies that reprogramming of fundamental miRNA and gene expression occurs both in barley and *P. indica*. Analysis of global changes in miRNA profiles of barley colonized with *P. indica* revealed that several putative endogenous barley miRNAs expressed upon colonization belonging to known micro RNA families involved in growth and developmental regulation.

## 1. Introduction

The interactions between plants and microorganisms, which are widely present in the microbial-dominated rhizosphere, have been well studied. This association is highly beneficial to the organisms involved, as plants benefit soil microorganisms by providing them with metabolites, while microorganisms promote plant growth and development by promoting nutrient uptake and/or protecting the plant from biotic and abiotic stresses [1,2]. The establishment and maintenance of mutualism requires genetic and epigenetic reprogramming and metabolomic regulation through the exchange of effector molecules between beneficial microorganisms and plants [3,4]. Beneficial microorganisms have a major role in crop production because of their impact on plant health and yield. *Piriformospora indica* (*P. indica*) is an endophytic fungus, belonging to the order Sebacinales that colonizes the roots of both monocotyledons and dicotyledons plants [5]. *P. indica* serves as an excellent model for beneficial microbes as it can form a mutually beneficial symbiosis with a series of crops such as Chinese cabbage, rice, wheat, cucumber, onion and banana, which can effectively promote their growth, nutrient absorption, accumulation of secondary metabolites, and resistance to disease damage [6,7,8,9,10,11,12]. *P. indica* has great potential in biological control and soil improvement and thus can play a positive role in agricultural production.

MicroRNAs (miRNAs) are a class of endogenous small noncoding RNAs (ncRNAs) which is evolutionary conserved and contains approximately 20–22 nucleotides [13,14]. They participate in the regulating gene expression and multiple physiological and biochemical processes by complementary functions with target gene mRNA. Many studies have shown that plant miRNAs play an important regulatory role in the interaction between plants and soil microbes, including promoting plant growth and development, stress response and hormone transduction [15]. MiRNAs recognize their mRNA target genes based on near-perfect complementarity and direct degradation or translational repression of homologous mRNA targets [16]. In addition, they tend to act as “early” regulators of signal transduction at the level of transcription factors (TFS) in various systems [17,18]. MiRNAs respond rapidly to infection by symbiotic bacteria. In soybean roots, a group of miRNAs which target a wide range of mRNAs were intensively up-or down-regulated by infection with the rhizobium bacterium *Brodyrnia japonicum* [19,20]. In the process of symbiosis, miRNAs were involved in the regulation of plant nutrient balance [21,22], hormone homeostasis and signal transduction [19], and spatial and temporal development of symbiosis nodules [20,23].

In addition, miRNAs also play a regulatory role in the process of abiotic stress in plants. Li et al. [24] found that after silencing BnmiR169n in rape seed, the drought tolerance of plants increased due to the increase of its target gene *BnNFYA8*. Under salt stress, the expression of miR169q in maize was inhibited and the expression of its target gene *ZmNFYA8* was upregulated. *ZmNFYA8* binds to the promoter of the antioxidant enzyme gene *zmper1* and activates its expression, alleviating the toxic effect of ROS on plants and improving maize salt tolerance [25]. The research on tomato by Zhang et al. [26] also verified that miR394 was involved in the negative regulation of biological stress. The overexpression of miR394 inhibited the expression of its target gene *LCR*, and then inhibited JA synthesis-related genes, thus reducing the resistance of tomato to *Phytophthora*. It is found that MiR164 plays an important role in wheat leaf rust and poplar black spot defense [27,28].

The role of miRNAs as gene expression regulators in Sebacinalean symbiosis has been largely unexplored. Previous studies have shown that *P. indica* can induce root growth of *Oncidium orchid* which is closely related to microRNA [29]. Results indicated that the predicted miRNAs target genes are mainly participated in auxin signal perception and transduction, transcription, development, and plant defense. Several novel unique miRNAs were detected, for which a function could not yet be identified. Another research revealed fundamental sRNA and gene expression reprogramming at the onset of symbiosis between *P. indica* and the model grass species *Brachypodium distachyon* [30]. Their data suggests that a Sebacinalean symbiosis involves reciprocal sRNA targeting of genes during the interaction.

Based on comprehensive high-throughput sequencing and transcriptome analysis, we evaluated an interaction between barley and the beneficial fungal root endophyte *P. indica* to elucidate miRNA regulatory changes in gene expression and miRNA–mRNA interaction profiles. Additionally, we discuss the biological functions and potential regulatory mechanisms of miRNAs in barley growth and possible miRNA-based regulation that might be crucial for the establishment of the barely-*P. indica* symbiosis. This study will contribute to our understanding of the RNA-based growth promotion mechanism mediated by *P. indica* colonization.

## 2. Methods

### 2.1. P. indica and Barley Cultivation and Inoculation

*Piriformospora indica* (11,827, Gift from IPAZ, Institute of Phytopathology, Giessen, Germany) was grown on complete media plates (CM [31]) at 23 °C in dark for one month. *P*. *indica* mycelium collection was carried out according to reference [32].

The seeds of barley line (gift from Tianjin Academy of Agricultural Sciences, JINNONG 8) were surface sterilized for 20 min with 3% active chlorine, sodium hypochlorite solution, and washed five times, then seeds were soaked in chlamydospore suspensions of *P. indica* (1 × 10^5^/mL) for 30 min. Control seeds were treated with Tween 20 (0.002%, *v*/*v*) solution as mock. The mock and soaking seeds (SS) were sown at the same time, with three replicates for each treatment. Barley biomass analyses were performed on seedlings grown on soil under 8 h dark (18 °C) and 16 h light (160 μmol m^−2^ s^−1^, 22 °C) conditions at 65% relative humidity for 4 weeks.

Samples for RNA-seq and RT-qPCR were also grown under these conditions. To assess growth promotion in *P. indica* inoculated barley relative to the control, we used the pairwise t test or the Mann–Whitney–Wilcoxon test on each of the three repetitions of experiments, after checking for normality and homogenous variances. Benjamini-Hochberg correction for multiple testing was used to correct the *p* values and the significance asterisks were assigned to the average *p*-value as follows: * for *p* ≤ 0.05, ** for *p* ≤ 0.001, and *** for *p* ≤ 0.0001.

### 2.2. Small RNA Library Construction and Sequencing

According to TRIzol (Invitrogen, #15,596-018) method, roots infected with *P. indica* at 3 and 7 days after inoculation (dai)or mock were collected, respectively. All samples were grounded into fine powder in liquid nitrogen for three independent repetitions. RNA degradation and contamination were monitored on 1% agarose gels. NanoPhotometer^®^ Spectrophotometer (IMPLEN, Carlsbad, CA, USA), Qubit^®^ RNA Assay Kit in Qubit^®^ 2.0 Flurometer (Life Technologies, Carlsbad, CA, USA) and RNA Nano 6000 Assay kit of Bioanalyzer 2100 system (Agilent Technologies, Santa Clara, CA, USA) were applied to check RNA purity, RNA concentration, and RNA integrity, respectively. Samples were subjected to small RNA Solexa sequencing in Novogene (Tianjin, China). Briefly, the small-molecule RNAs were separated by 15% (*w*/*v*) PAGE (18–30 nt), and then the purified small molecule RNAs were ligated to a pair of Solexa adaptors to the 59 and 39 ends, reverse transcribed to cDNA using a RT primer, and finally amplified by PCR and sequenced. The RNA-seq data have been submitted to the Sequence Read Archive (SRA) on NCBI website, and the accession number is PRJNA898289.

### 2.3. Novel miRNA Prediction

The characteristics of hairpin structure of miRNA precursor could be used to predict novel miRNA. We used miREvo [33] and mirdeep2 [34] to predict novel miRNAs and the minimum free energy of the small RNA tags unannotated in the former steps. At the same time, custom scripts were used to obtain the identified miRNA counts as well as base bias on the first position with certain length and on each position of all identified miRNAs, respectively.

### 2.4. Co-Expression Analysis of mRNA-miRNA

The functional annotation of identified miRNAs was performed using co-expression analysis [35]. Pearson’s correlation coefficients between mRNAs and miRNAs were calculated based on the mRNAs FPKM values, and the putative target mRNA should have a value >0.99 or <−0.99. In addition, the TargetFinder [36] was used to predict the target mRNA of the miRNA. The mRNA–miRNA network was constructed using Cytoscape [37] software (Version 3.0.2) based on the correlations between mRNAs and miRNAs.

### 2.5. GO and KEGG Enrichment Analysis of Differentially Expressed Genes

Gene Ontology (GO) enrichment analysis of differentially expressed genes was implemented by the cluster Profiler R package, in which gene length bias was corrected. GO terms with corrected *p*-value less than 0.05 were considered significantly enriched by differential expressed genes. The Kyoto Encyclopedia of Genes and Genomes (KEGG) is a database resource for understanding high-level functions and utilities of the biological system, such as the cell, the organism, and the ecosystem, from molecular-level information, especially large-scale molecular data sets generated by genome sequencing and other high-throughput experimental technologies (http://www.genome.jp/kegg/ (accessed on 12 November 2022)). We used cluster Profiler R package and KOBAS software to test the statistical enrichment of differential expression genes. The term with a corrected *p*-value < 0.05 is considered to be significantly enriched in differentially expressed genes.

### 2.6. Quantitative Real Time Polymerase Chain Reaction (qRT-PCR) and Stem-Loop PCR for Validation of Sequencing Results

The RNA collected in 2.2 and M-MLV Reverse Transcriptase were used to make cDNA. After treatment with Dnase I (Sigma, Germany), the cDNA was used as a template for qRT-PCR to quantify selected miRNAs and mRNAs using the miRNA-specific primers and target mRNA specific primers [38]. The expression level of respective gene was determined by quantitative RT-PCR. Quantitative RT-PCR was measured by SYBR Green influorescence method as described previously. In brief, qPCR experiments were conducted on a Light Cycler96 Fast real-time PCR system (Roche). The reaction solution contains 2 × Ultra SYBR Mixture 10 µL, 100 ng cDNA template, 10 µM forward and reverse primers. HvUBIQUTIN was used as the control, and all experiments were conducted with at least three technical replications. The amplification program was applied as the following steps: the first initial activation step was performed at 95 °C, 5 min, then followed by 30 cycles (95 °C for 20 s, 56 °C for 35 s, 72 °C for 35 s, and 65 °C for 20 s). At the end of each cycle, melting curves were determined respectively to guarantee the amplification of the single-PCR product.

For the identification of miRNAs in barley, stem-loop RT-PCR was referenced [39].

Briefly, cDNA was synthesized from total RNA extracted from *P. indica* inoculated barley roots. Hairpin primer was designed and performed according to literature [40]. For each stem-loop reaction, the detail protocol was performed according to the manufacturer’s instructions (Thermo Scientific, Waltham, MA, USA).

For primer annealing, the reaction was incubated at 16 °C for 30 min and then extended at 42 °C for 30 min. The universal stem-loop primer and specific miRNA primer (Appendix A) were used in Endpoint PCR under the same conditions as described in target RNA amplification. PCR products were purified and cloned into the pGEM^®^-T Vector Systems (Promega, Madison, WI, USA) following the manufacturer’s instructions. Five colonies of each cloned miRNA were subjected to sequencing using an M13 forward primer (Ding guo, China).

## 3. Results

### 3.1. P. indica Promote Root Growth and Plant Development

To investigate whether barley seed can develop a beneficial interaction with *P. indica* by applying seed soaking (SS) treatment with *P. indica* chlamydospore suspensions, seedling three days after germination in soil were subjected to colonization identification. The hyphae of *P. indica* were widely distributed over the root surface three days after germination (Figure 1A) which indicating that the establishment of the beneficial symbiosis is successful. The biomass enhancement was observed in the SS treatment compared to that the non-inoculated barley, the stems and roots developed better than barley without *P. indica* co-cultivation at 3 and 7 days after inoculation (dai) (Figure 1B,C). Furthermore, the branching of roots was also obvious (Figure 1D). Comparison of shoot length in colonized vs. non-colonized plants grown in soil showed that *P. indica* increased the shoot length by 18.7% and 22.1% at 3 dai and 7 dai, respectively (Figure 1E), and total grain weight/plant increased by 36.9% and 44.6% at 3 dai and 7 dai, respectively (Figure 1F). We found that this method of soaking seeds was very effective in increasing barley biomass. Concordantly, shoot and weight analyses of barley seedlings revealed a significant increase in biomass (Figure 1D) upon *P. indica* colonization.

### 3.2. Establishment of the Barley–P. indica Interaction Is Associated with Extensive Transcriptional Reprogramming

In order to evaluate how the mutualistic interaction affects, the miRNA profiles in the colonized barley root at 3 and 7 dai of *P. indica* and barley controls were subjected to miRNA sequencing. These time points were selected because the fungus-related growth-promoting effect was visible. After removal of the low-quality contaminant and adapter reads, 7,798,928, 6,418,039, and 7,136,192 clean read sequences were obtained from the libraries of mock, 3 dai, and 7 dai roots (Appendix A), respectively. Sequencing data indicated that libraries are of high quality and can be used for further miRNA studies (Appendix A).

Differentially expressed microRNAs in barley were found in the three comparison groups, including 3 dai vs. mock, 7 dai vs. mock, and 7 dai vs. 3 dai (Figure 2). There were 8 microRNAs upregulated and 3 micro RNAs down regulated in 3 dai vs. mock; there were 11 microRNAs upregulated and 13 micro RNAs down regulated in 7 dai vs. mock, and there were 3 microRNAs upregulated and 4 microRNAs down regulated in 7 dai vs. 3 dai. The clustering heat map showed that *P. indica* colonization affected the expression pattern of microRNAs (Appendix A); alternatively, some miRNAs like novel-1, miR-444b, miR-1120 and miR397a were down regulated responding to the *P. indica* colonization; whereas miR171-3P and miR 6183 were upregulated at 3 dai and down-regulated at 7 dai; the other microRNAs including miR1681, novel-22, novel-3, miR168-3P, miR6198, miR6177, and novel 13 were up-regulated after *P. indica* colonization.

To investigate interaction of barley by *P. indica*, differentially expressed genes (DEGs) in colonized *P. indica* in comparison to axenic mycelium samples were analyzed. Among the 667 DEGs of predicted unique *P. indica* genes, 499 were confirmed as up-regulated and 168 were down-regulated in 7 dai vs. control, and of 594 predicted unique *P. indica* genes, 517 were confirmed as upregulated and 168 were down-regulated at day 3 after colonization (Figure 3). Gene ontology and annotation analysis indicated that the DEGs with the greatest changes were mainly concentrated in oxidoreductase activity, ion transmembrane transporter activity, phosphate transporter and tRNA THr modification in 3 dai vs. control, and other DEGs were concentrated in cellular respiration, ion transport, endonuclease activity, oxidoreductase activity, and phosphate transporter in 7 dai vs. control (Figure 4). Top 20 *P. indica* DEGs during colonization at 3 dai and 7 dai vs. mock and 7 dai vs. 3 dai are listed in Table 1, Table 2 and Table 3.

### 3.3. Differentially Expressed miRNAs

Analysis of unique plant miRNAs in 7 dai-mock vs. 3 dai-mock revealed that 12 of the putative endogenous miRNAs were exclusively present in 7 dai-mock, 6 miRNAs were exclusively present in 3 dai-mock, and 12 miRNAs were present in both comparison group (Figure 5). For the reads from the putative 3 dai-mock vs. 7 dai-3 dai, 13 miRNAs were exclusively present in 3 dai-mock, 4 miRNAs were exclusive present to 7 dai-3 dai, and 5 miRNAs were found in both comparison groups. Comparison between the unique miRNAs in 7 dai-mock vs. 7 dai-3 dai indicated that 19 of the putative endogenous miRNAs were exclusively present in 7 dai vs. mock, 4 miRNAs were found in 7 dai vs. 3 dai, and 5 miRNAs were present in both comparison group. Similarly, from the putative 3 dai-mock vs. 7 dai-3 dai, three miRNAs were exclusively present in 3 dai-mock, nine miRNAs were exclusively present in 7 dai-mock, one miRNA was exclusively present in 7 dai-3 dai, and two miRNAs were found in all three comparison groups.

### 3.4. Expression Patterns of miRNAs and Their Putative Targets in P. indica—Colonized Roots

To elucidate the regulatory function of miRNAs on their putative targets, real-time qPCR was performed to confirm the expression level by using specific primers for these miRNAs and their target genes (Appendix A). The miRNAs novel_1, novel_22, novel_40, miR444b and their target genes involved in the growth-regulating factor, promoter-binding-like protein and transcription factor were selected. QPCR results indicated that the selected miRNAs novel_1 was down-regulated at 3 dai compared with mock, and its target genes HORVU7Hr1G012380. 5 and HORVU5Hr1G055920. 2 were both down-regulated, too (Figure 6A). In addition, the selected miRNAs miR444b was down-regulated at 3 dai compared to mock, and its target gene HORVU3Hr1G076030. 14 was up-regulated, but the other target gene HORVU1Hr1G006020. 8 was down-regulated (Figure 6B). Moreover, the selected miRNAs novel_22 was up-regulated at 3 dai compared with mock, and its target gene HORVU4Hr1G082910. 18 was upregulated, but the target gene HORVU2Hr1G085210. 5 was downregulated (Figure 6C). And the selected miRNAs novel_40 was down-regulated at 3 dai compared to mock, and its target gene HORVU5Hr1G054420. 4 was up-regulated, another target gene HORVU7Hr1G088630. 3 was down-regulated (Figure 6D). The result showed that the expression of these miRNAs was consistent with the results of RNA-seq data.

Functions of target mRNA corresponding to up-regulated and down-regulated miRNA in 3 dai vs. mock are listed in Table 4 and Table 5. As shown, one miRNA corresponded to multiple target genes, and the regulatory function of miRNA on target genes may be either up-regulated or down-regulated. Data showed that four microRNAs corresponding to target RNAs had specific functional descriptions. And the functions of these target mRNA described as: Serine/threonine-protein kinase STY13, putative transcription factor RL9, leucine rich repeat family expressed, 60S ribosomal protein L35a-1, squamosa promoter-binding-like protein, squamosa promoter-binding-like protein 4/16, scarecrow-like protein 6/. And the functions of down-regulated miRNA related target mRNA were MADS-box transcription factor 57, 6-phosphogluconate dehydrogenase, general negative regulator of transcription subunit 3 isoform X4, and growth-regulating factor 1, 2, 4, 5-like isoform.

### 3.5. Prediction of miRNA Target Genes

To understand the regulatory function of miRNAs during symbiosis, those miRNAs which were abundantly detected and significantly up-/down-regulated by *P. indica* were selected for further investigation. Eighty-one conserved miRNAs belonging to 15 families were selected for target gene prediction. Target-Finder software was used to predict miRNA target genes, 450 best fit target candidates were obtained (Appendix A).

Subsequently, annotation and GO analysis were conducted by Goseq. A total of 450 targets were annotated and distributed in 38 categories. Clustering of microRNA target genes varied among different comparison groups. MiRNAs target genes were more concentrated in biological process signaling pathways than in cell components in 3 dai vs. mock (Figure 7A): In the subcategory of biological process, most of the target genes of miRNA are concentrated in the cellular biosynthetic process, macromolecule biosynthetic process and nucleic acid metabolic process; In the subcategory of molecular function, transferase activity and copper ion binding are the main enrichment pathways of miRNA target genes. In the 7 dai vs. mock group, miRNA target genes were involved in biological processes, cell components, and molecular functional signaling pathways (Figure 7B): most of the miRNA target genes are concentrated in the cellular biosynthetic process, macromolecule biosynthetic process, nucleic acid metabolic process and organic substance metabolic process in the subcategory of biological process, In the subcategory of cell components, most of the miRNA target genes are concentrated in intracellular organelle, membrane bounded organelle and nucleus; in the subcategory of molecular function, transferase activity, ribonucleotide binding, carbohydrate derivative binding, and anion binding are the main enrichment pathways of microRNA target genes. In the 7 dai vs. 3 dai group, miRNAs target genes were more concentrated in molecular function signaling pathways than in biological process (Figure 7C). In the subcategory of biological process, the expression of miRNA target genes was mainly enriched in protein phosphorylation and phosphorous compound metabolism. Whereas, miRNA target genes, in the subcategory of molecular function were mainly concentrated in binding process of small molecules, ATP, ADP, copper ion, carbohydrate derivative, and purine nucleoside.

### 3.6. Identification of miRNAs Related to Transcription Factor and Other Key Pathway Regulation in Barley Roots Colonization by P. indica

To further study the function of miRNAs in the growth of barley roots in response to *P. indica* colonization, the GO terms of different expression miRNAs targets were annotated based on http://geneontology.org/ (accessed on 12 November 2022) and http://www.uniprot.org/uniprot/ (accessed on 12 November 2022). The results showed that seven miRNAs might be involved in the regulation of gene transcription, because their target genes have transcription factor activity (Table 6). In the 3 dai vs. mock group, miR6189 was up-regulated and miR 6214 and miR444b were down-regulated; in the 7 dai vs. mock group, miR6190 was up-regulated and miR6214, miR397a were down-regulated; in the 7 dai vs. 3 dai group, miR6180 was down-regulated and miR6214 was up-regulated; all their target genes have transcription factor activity.

Besides the transcription factor activity regulation, we found that miRNAs including miR6214, miR6180, miR6189, miR444b, miR6190, and miR397a might also be involved in other key signal pathways because their target genes participated in these pathways including cell division, auxin stimulus, photosynthesis, hormone stimulus, and chlorophyllide oxygenase activity, which contribute to the plant growth promotion. It was observed that there were four miRNAs response to auxin stimulus in the 3 dai vs. mock and 7 dai vs. mock comparison groups. Those miRNAs were all up-regulated compared with mock while they were diversely (positively or negatively) correlated to their target genes (Table 7).

### 3.7. MiRNA–mRNA Interaction

MiRNA up-regulated and down-regulated its expression in response to the colonization of *P. indica*, thereby affecting the expression of plant transcriptome. The miRNA–mRNA interaction diagrams of 3 dai vs. mock and 7 dai vs. mock were obtained by Cytoscape software, as shown in Figure 8A,B. Several key miRNAs including novel_1, novel_3, novel_22, novel_40, miR444b, miR6181, miR6198 were selected to construct the miRNA–mRNA interaction network. One miRNA can establish interactions with multiple target genes. Novel_1 showed strong positive correlation (r = 0.7852) with its targets HORVU7HrG012380. 5, HORVU7HrG055920. 2, HORVU4HrG050160, and HORVU4HrG064020 because both miRNA and its target genes were down-regulated. As shown in Table 2 and Table 3, most target mRNAs of novel_1 were involved in growth-regulating factor. It is reasonable to speculate that novel_1 acted as a trans-regulator regulating growth regulator in response to *P. indica* colonization. In contrast, miR444b showed a strong negative correlation (r = 0.7013) with its target mRNAs including HORVU3HrG076030, HORVU2HrG079610, HORVU5HrG094420.6, HORVU5HrG094420. 8, HORVU5HrG094420. 5, and HORVU4HrG087340. 8., except HORVU1HrG006020. Those target mRNAs of novel_444b were involved in MADS-box transcription factor 57.

## 4. Discussion

To cope with rapidly changing environments, plants employ a large number of mechanisms that provide phenotypic plasticity and allow fine-tuning of stress response actions. Advances in molecular biology have made great strides in identifying genomic regions and underlying mechanisms that influence transcriptional and post-transcriptional biotic and abiotic stress pathways regulation. In plants, miRNAs evolve and contribute to the complexity of these molecules through a series of pathways, and play an important role in the regulation of stress response activity. It has been shown that one way in which plants respond to environmental stress is through the activity of miRNAs. MiRNAs, as important regulatory molecules of plant biotic and abiotic stress response, are the driver molecules of RNA interference (RNAi), and ensure the up-regulation and down-regulation of target genes, and participate in important biological processes [41]. MiRNAs are small, 20–22 nt noncoding RNAs that regulate gene expression by post-transcriptional gene silencing in most eukaryotes [42]. RNA interference (RNAi) regulates gene expression by inducing degradation of messenger RNA (mRNA) or inhibiting its translation. MiRNAs play crucial roles in plant development, including the formation of embryo, meristem, leaf, and flower [43] as well as the responses to biotic and abiotic stresses [44].

*P. indica* is well known to be able to establish beneficial interactions with many different hosts, including monocotyledons such as barley, wheat, rice, corn, and dicotyledons such as *Arabidopsis* and tobacco [8], even on Brassicaceae family that cannot be colonized by mycorrhizal fungi [11]. Colonization of the roots by *P. indica* results in enhanced biomass production as well as increased resistance against biotic and abiotic stresses. We established and studied the interaction between barley—a major cereal crop—and *P. indica*—a beneficial endophyte with an exceptionally large host range. The functions and regulatory mechanisms of miRNAs in barley growth regulation during the symbiosis with *P. indica* were explored in our work. We showed that *P. indica* colonizes barley, resulting growth promotion in shoot, alterations in root architecture, and improved grain development.

### 4.1. Transcriptional Changes Detected during the Barley–P. indica Interaction

To investigate the interaction of barley with *P. indica*, we analyzed DEGs in colonized *P. indica* in comparison to axenic mycelium samples. Gene ontology analysis indicated DEGs with the greatest changes were mainly concentrated in oxidoreductase activity, ion transmembrane transporter activity, phosphate transporter, and tRNA THr modification in 3 dai vs. control, and other DEGs were concentrated in cellular respiration, ion transport, endonuclease activity, oxidoreductase activity and phosphate transporter in 7 dai vs. control. We noticed that the phosphate transporter gene which could promote the uptake of phosphorus by plant roots and provide essential nutrients for plants [45] was upregulated in both 3 dai vs. control and 7 dai vs. control. These results indicate that the interaction between barley and *P. indica* initiates the expression of proteins related to phosphorus transport. Other up-regulated genes encode enzymes involved in ion transport and ion transmembrane transporter activity. These proteins may modulate metal ions transport from soil into the plants. In fact, there are many transporters in *P. indica* which are responsible for the transfer of phosphorus [45] and sulfur elements [46] and the transport of metal elements. It is believed that these transporters play an important role in assisting the plant to absorb large and trace elements.

Roots of barley plants also displayed substantial transcriptional reprogramming following *P. indica* colonization. GO analysis indicated DEGs in barley root enrichment in nucleic acid metabolic and macromolecule biosynthetic, cellular marcomolecule biosynthetic and transferase activity associated processes in 3 dai vs. mock. Barley root DEGs exhibiting the greatest changes in expression between 7 dai and noncolonized plants are related to the primary metabolic process, cellular process, organic substance, transferase activity, and phosphotransferase activity. Previous research indicated that DEGs in *Brachypodium distachyon* after *P. indica* colonization involved in catalytic and oxidoreduction associated processes [30]. This also indicates that different plants have different transcriptome responses upon *P. indica* colonization. In 3 dai vs. mock, of the downregulated barley genes, several encode proteins commonly associated with stress responses, including a peroxidase and a putative protein kinase. Several downregulated DEGs also encode transcription factors, including growth-regulating factor 1/2/4/5/6-like (Appendix A). One report also demonstrated that several DEGs encode transcription factors, including MYB-related, GRAS, and bZIP were downregulated in *Brachypodium distachyon* after *P. indica* colonization. Of the upregulated barley genes, several encode leucine rich repeat family and squamosa promoter-binding-like protein (Appendix A). In 7 dai vs. mock, several upregulated DEGs also encode squamosa promoter-binding-like protein 16, serine/threonine-protein kinase SIS8, and some down regulated DEGs encode MADS-box transcription factor 57, pyruvate kinase, cytosolic, isozyme, and transcription initiation factor TFIID subunit 6 (Appendix A). In 7 dai vs. 3 dai, the downregulated target genes were mainly encoded scarecrow-like protein and growth-regulating factor (Appendix A).

### 4.2. Barley miRNAs Detected in the Barley–P. indica Interaction

The role of miRNAs as gene expression regulators in *P. indica* symbiosis has been largely unexplored in barley. The report demonstrated that *P. indica* promot plant growth associated miRNAs in *Oncidium orchid* roots [29]. Another study indicated that sRNAs reprogrammed after *P. indica* colonization in *Brachypodium distachyon* [30]. A high-throughput sequencing and comparative expression analysis were conducted. In total, 7,798,928, 6,418,039 and 7,136,192 clean reads were obtained from the libraries in control and *P. indica*-colonized root libraries. Differentially expressed microRNAs were found in the three comparison groups, including 3 dai vs. mock, 7 dai vs. mock and 7 dai vs. 3 dai (Figure 2). Analysis of putative endogenous barley miRNAs expressed during *P. indica* colonization identified 42 miRNAs. Some of them have unknown targets, whereas some of them have more than one target. For example, the miRNA hvu-miR1120 targets mRNAs (HORVU1Hr1G080480) encoding 6-phosphogluconate dehydrogenase and other targets mRNAs (HORVU4Hr1G002170) encoding general negative regulators of transcription. Four key miRNAs, including hvu-miR6181, novel_22, novel_44, and hvu-miR6198, were up-regulated both in 3 dai and 7 dai vs. mock in our work. Their targets gene functions were mainly involved in vacuolar protein-sorting-associated protein and squamosa promoter-binding-like protein. Other important miRNAs, such as hvu-miR1120, hvu-miR444b, novel_1, and hvu-miR397a, were down-regulated both in 3 dai and 7 dai vs. mock. Among them, hvu-miR6189, hvu-miR6214, hvu-miR444b, hvu-miR6190, hvu-miR397a, their targets mRNAs actually possess transcription factor activity (Table 4). In *Arabidopsis*, repression of transcription factors by the miR165/166 family modulates root growth, maintenance of the shoot apicalmeristem, and the development of leaf polarity; miR156-mediated downregulation of SPLs modulates developmental timing, lateral root development, branching, and leaf morphology [30]. This suggests that miRNAs, which regulate transcription factors, play an important role in plant growth and development. MiRNA including mir156, mir166, and mir169, which target mRNAs for the transcription factor genes SPL, PHV/PHB, and NTF, were also abundantly detected in *Oncidium orchid* roots after *P. indica* colonization [29]. This further indicates that the colonization of *P. indica* could reprogram miRNAs which participate in transcription factors regulation.

These miRNAs identified in barley include hvu-miR6180, hvu-miR6189, hvu-miR6214, hvu-miR444b, and hvu-miR6190, predicted to target genes involved in hormone activity, cell division, and photosynthesis pathway (Table 5). Because these miRNAs predict a variety of targets that are associated with plant growth and development, this group of miRNAs may play an important role in reprogramming plant cells during *P. indica* symbiosis establishment.

### 4.3. MiRNA and Target mRNA

MiRNAs regulate gene expression by mediating target gene silencing at transcriptional (TGS) and post-transcriptional (PTGS) levels, including DNA methylation, histone modification, translational repression, and RNA silencing [47,48]. They play important roles in plant development, differentiation, and response to biotic and abiotic stresses [48,49,50,51,52,53,54,55]. In the analysis of the interaction between miRNA and target genes, it was found that miRNA can be positively correlated with target genes or negatively correlated with target genes at 3 dai and 7 dai of *P. indica*. The predicted target genes of these miRNAs are mainly involved in transcription, cell division, auxin signal perception and transduction, photosynthesis, and hormone stimulus (Table 4 and Table 5). In fact, miRNAs can bind not only to mRNAs but also to long non-coding RNAs (lncRNA). LncRNAs acting as potential competing endogenous RNA-harboring microRNA response elements (MREs), thereby competing with mRNAs for shared miRNA and thus regulate miRNA-mediated gene silencing [44]. In our work, the lncRNA–mRNA–miRNA network was constructed, too (Appendix A). For instance, several key lncRNAs were found to have correlations with the miRNA including novel_1, novel_19, novel_22, novel_44hvu-miR6181, hvu-miR6198, and hvu-miR444b in 7 dai vs. mock; several key lncRNAs were also found to have correlations with the miRNAs, including novel_1, novel_2, novel_22, hvu-miR6181, hvu-miR6198, and hvu-miR444b in 3 dai vs. mock. These data suggest that these key miRNAs excavated play an important role in the regulation of plant growth in response to *P. indica* colonization. Specific regulatory functions of miRNAs on target genes and lncRNAs is the main research content in the future work.

## 5. Conclusions

We reported the miRNA profiling of barley after colonization by *P. indica*. The miRNAs and their target genes illustrated that the physiological metabolism of barley is reprogrammed in response to the symbiotic interaction. Genes participating in transcription, cell division, auxin signal perception and transduction, photosynthesis, and hormone stimulus are major targets of the *P. indica*-induced miRNAs in barley roots. Therefore, we propose that *P. indica* alters the miRNA pattern to establish an intricate network for growth promotion and developmental reprogramming and enhances resistance in barley roots. Several novel unique miRNAs were detected, for which a function could not yet be identified. Further investigations on the molecular mechanism of miRNAs in symbiotic interactions are of huge significance.

## Figures and Tables

**Figure 1 jof-09-00024-f001:**
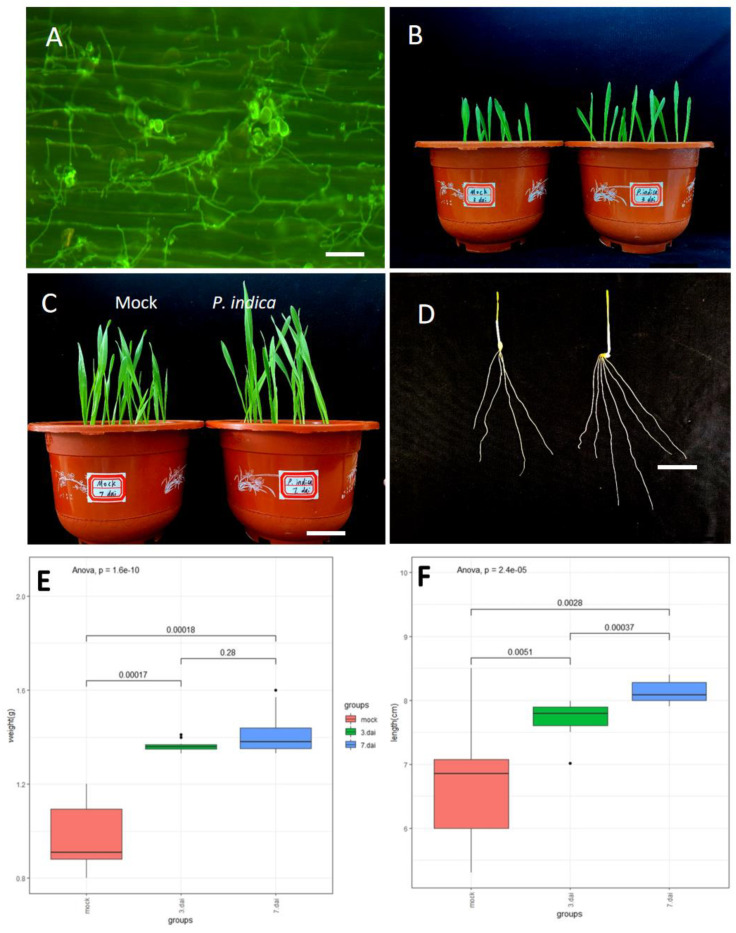
Root colonization by *P. indica* increases the growth and yield of barley. (**A**) Colonization pattern of *P. indica* on barley roots. Fluorescence microscopy showing WGA-AF488 staining of *P. indica* cell walls (λexc494 nm, λem515). (**B**,**C**) Total grain biomass of control vs. colonized plants at 3 dai and 7 dai. (**D**) Root branching between control and colonized roots at 3 dai. For b and c, barely seeds were seed soaking with 5 × 10^5^ chlamydospores per ml and grown in soil. (**E**) Shoot length of 3 dai vs. mock and 7 dai vs. mock. (**F**) weight of the seedlings from 3 dai vs. mock and 7 dai vs. mock Sample size *n* = 10. The results are from three independent biological replicates. The significance threshold for *p* values, corrected for multiple testing (Benjamini–Hochberg) was set at 0.05.

**Figure 2 jof-09-00024-f002:**
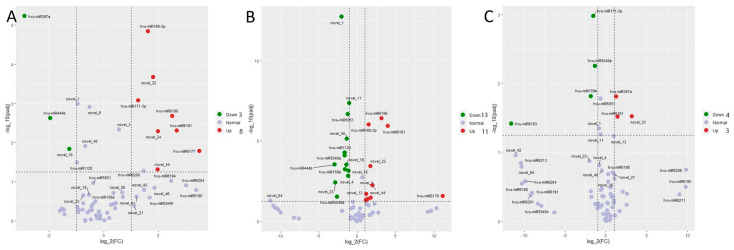
Volcano plots of colonization-associated, differentially expressed miRNA in barley. (**A**) Barley differentially expressed miRNA identified by comparing reads from colonized roots at 3 dai vs. mock. (**B**) Barley differentially expressed miRNA identified by comparing reads from colonized roots at 7 dai vs. mock. (**C**) Barley differentially expressed miRNA identified by comparing reads from colonized roots at 7 dai vs. 3 dai. The dashed line in the figure shows the set threshold, vertical line (x = ±1) and horizontal line (y = 2). Genes with padj values less than 0.05 and absolute values of foldchange greater than 2 are generally considered as differential genes.

**Figure 3 jof-09-00024-f003:**
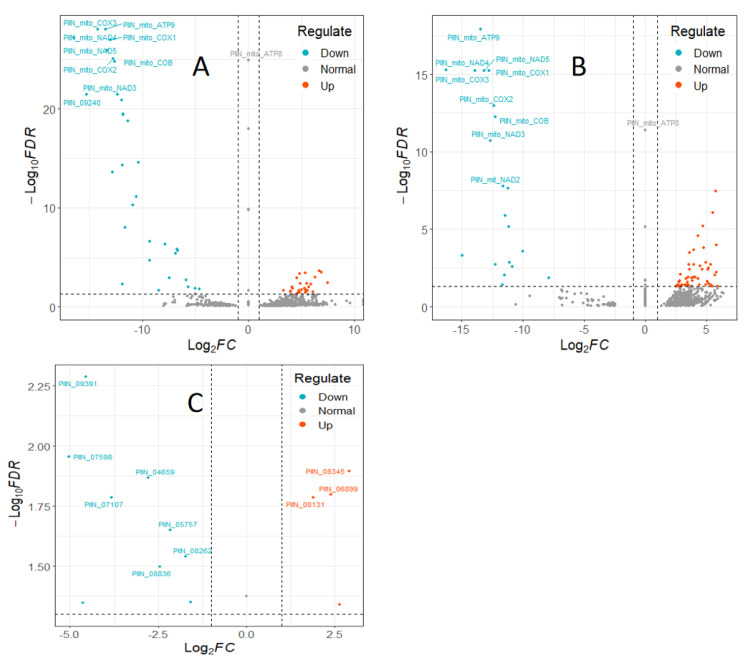
Volcano plots of colonization-associated, differentially expressed genes (DEGs) in *P. indica.* (**A**) DEGs in *P. indica* identified by comparing reads from colonized roots at 3 dai vs. axenic mycelium. (**B**) DEGs in *P. indica* identified by comparing reads from colonized roots at 7 dai vs. axenic mycelium. (**C**) *P. indica* DEGs identified by comparing reads from colonized roots at 7 dai vs. 3 dai. The dashed line in the figure shows the set threshold, vertical line (x = ±1) and horizontal line (y = 2). Genes with padj values less than 0.05 and absolute values of foldchange greater than 2 are generally considered as differential genes.

**Figure 4 jof-09-00024-f004:**
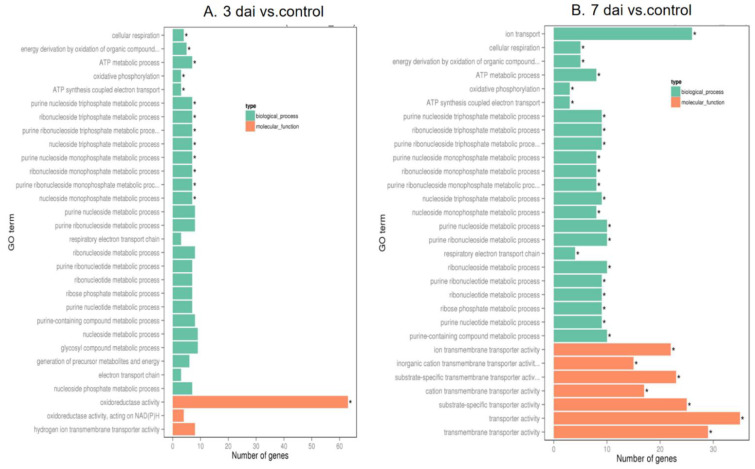
Go enrichment histogram of differentially expressed genes (DEGs) in *P. indica*. (**A**) GO enrichment in 3 days vs. control; (**B**) GO enrichment in 7 days vs. control. The abscissa denotes the name of GO entry, which is divided into three categories by box: BP: biological process, CC: cell component, MF: molecular function, distinguished by different frames, and the ordinate is the number of genes enriched by GO entry. The significance threshold for p values, corrected for multiple testing (Benjamini–Hochberg) was set at 0.05 (* ≤ 0.05).

**Figure 5 jof-09-00024-f005:**
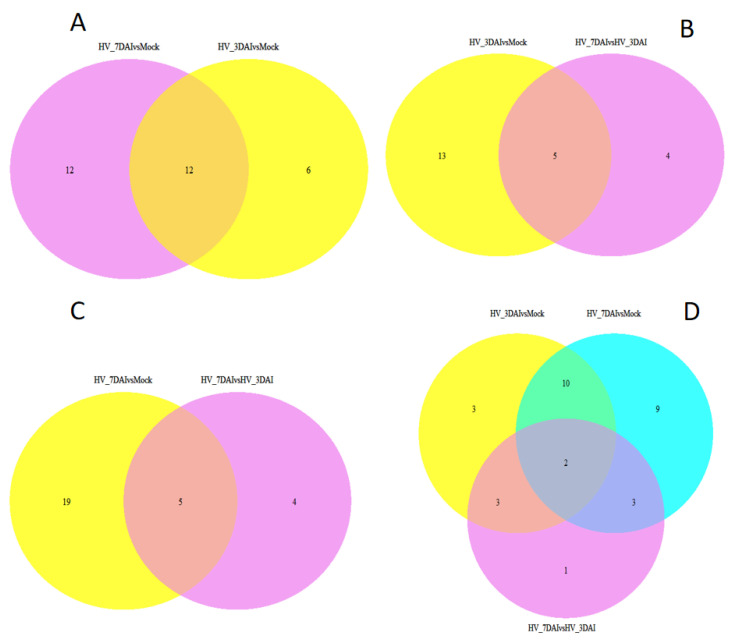
Venn diagrams showing the sample-exclusive or communal presence of unique putative endogenous miRNAs. (**A**) putative endogenous miRNAs in 7 dai-mock vs. 3 dai-mock, (**B**) miRNAs in 3 dai-mock vs. 7dai-3dai; (**C**) miRNAs in 7 dai-mock vs. 7 dai-3 dai; (**D**) miRNAs in 3 dai-mock vs. 7 dai-3 dai vs. 7 dai-mock.

**Figure 6 jof-09-00024-f006:**
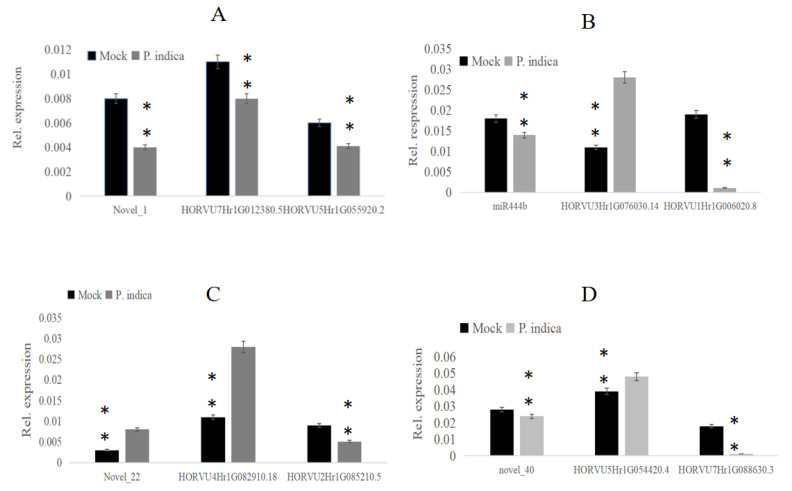
Go enrichment histogram of differentially expressed genes (DEGs) in barley. (**A**) GO enrichment in 3 dai vs. mock; (**B**) GO enrichment in 7 dai vs. mock. (**C**) GO enrichment in 7 dai vs. 3 dai. The abscissa denotes the name of GO entry, which is divided into three categories by box: BP: biological process, CC: cell component, MF: molecular function, distinguished by different frames, and the ordinate is the number of genes enriched by GO entry. (**D**) The chart shows that miRNA_novel 40 was downregulated and its target genes HORVU5Hr1G054420.4 was upregulated; another target gene HORVU7Hr1G088630 was downregulated. The experiments of q-PCR and the data analyses were performed in three biological replicates. ** Highly significant difference (*p* < 0.01).

**Figure 7 jof-09-00024-f007:**
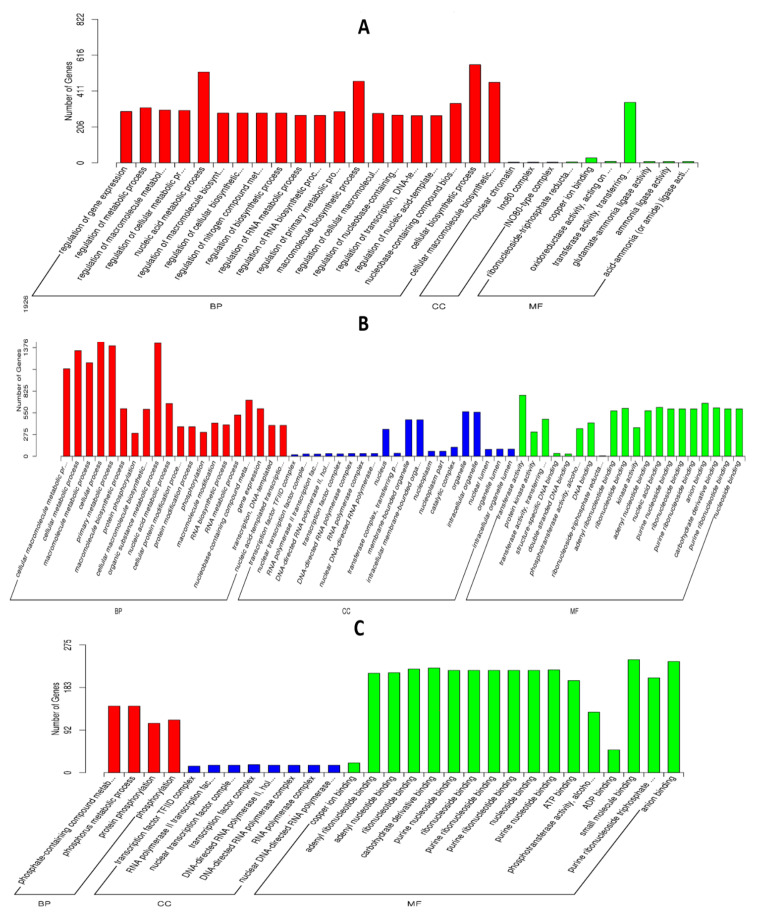
The relative expression of miRNAs and their target genes involved in the growth-regulating factor, promoter-binding-like protein, and transcription factor. (**A**) The chart shows that miRNA_novel 1 was downregulated and its target genes HORVU7Hr1G012380 and HORVU5Hr1G055920 were down-regulated. (**B**) The chart shows that miRNA_444b was downregulated and its target genes HORVU3Hr1G076030 was upregulated but HORVU5Hr1G006020.8 were down-regulated. (**C**) The chart shows that miRNA_novel 22 was upregulated and its target genes HORVU4Hr1G082910 was upregulated, too. The other target gene HORVU2Hr1G085210 was downregulated.

**Figure 8 jof-09-00024-f008:**
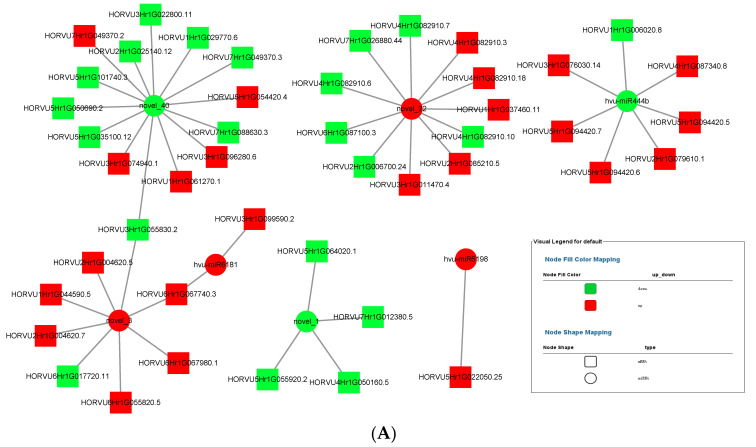
Networks of miRNA–mRNA. (**A**) Network of miRNA–mRNA for 3 dai vs. mock. miRNA–mRNA network was constructed for 3 dai vs. mock based on the co-expression correlation of miRNA–mRNA (Circle node: miRNA. Diamond nodes: mRNA. Red color: up-regulation; green color: down-regulation). (**B**) Network of miRNA-mRNA for 7 dai vs. mock. miRNA–mRNA network was constructed for 7 dai vs. mock based on the co-expression correlation of miRNA-mRNA (Circle node: miRNA. Diamond nodes: mRNA. Red color: up-regulation; green color: down-regulation).

**Table 1 jof-09-00024-t001:** Top 20 *P. indica* differentially expressed genes (DEGs) during colonization (3 dai vs. control).

Gene	Description	Log2FC	EXP
PIIN_mito_NAD4	oxidoreductase activity	16.468	up
PIIN_mito_NAD2	oxidoreductase activity	11.841	up
PIIN_mito_COX3	membrane	14.225	up
PIIN_mito_ATP9	ion transport	13.467	up
PIIN_mito_NAD5	oxidoreductase activity	13.356	up
PIIN_mito_COX1	Ion transmembrane transporter activity	13.053	up
PIIN_mito_NAD3	oxidoreductase activity	12.363	up
PIIN_mito_COX2	ion transmembrane transporter activity	12.764	up
PIIN_mito_COB	membrane	12.641	up
PIIN_mit_GIY1	molecular_function	12.236	up
PIIN_mit_LAG4	endonuclease activity	11.708	up
PIIN_mit_NAD2	oxidoreductase activity	11.65	up
PIIN_08131	phosphate transporter	8.031	up
PIIN_mit_LAG8 PIIN_mito_NAD6 PIIN_mito_NAD1 PIIN_mit_LAG7 PIIN_mito_ATP6 PIIN_05864 PIIN_03862 PIIN_mito_RPS3	None oxidoreductase activity membrane endonuclease activity ion transport ion transport tRNA THr modification None	11.505 11.444 11.201 11.166 11.087 10.883 10.583 10.045	up up up up up up up up

**Table 2 jof-09-00024-t002:** Top 20 *P. indica* differentially expressed genes (DEGs) during colonization (7 dai vs. control).

Gene	Description	FC	EXP
PIIN_mito_NAD4	cellular respiration	16.297	up
PIIN_09240	None	14.972	up
PIIN_mito_COX3	membrane	13.926	up
PIIN_mito_ATP9	ion transport	13.449	up
PIIN_mito_NAD5	cellular respiration	13.182	up
PIIN_mito_COX1	Ion transmembrane transporter activity	12.757	up
PIIN_mito_NAD3	oxidoreductase activity	12.645	up
PIIN_08131	phosphate transporter	12.531	up
PIIN_mito_COX2	ion transmembrane transporter activity	12.391	up
PIIN_mito_COB	membrane	12.268	up
PIIN_mito_NAD3 PIIN_mito_ATP6 PIIN_mit_GIY1 PIIN_03862 PIIN_mito_NAD6 PIIN_mit_NAD2 PIIN_mit_LAG7 PIIN_mito_NAD1 PIIN_mit_LAG8 PIIN_05864 PIIN_mito_RPS3	oxidoreductase activity ion transport molecular_function tRNA THr modification oxidoreductase activity oxidoreductase activity endonuclease activity membrane None ion transport None	12.363 11.975 11.899 11.875 11.858 11.841 11.62 11.379 10.883 10.596 10.353	up up up up up up up up up up up

**Table 3 jof-09-00024-t003:** *P. indica* differentially expressed genes (DEGs) during colonization (7 dai vs. 3 dai).

Gene	Description	FC	EXP
PIIN_09391	copper ion transmembrane transporter activity	4.543	up
PIIN_07598	serine-type endopeptidase activity	5.0228	up
PIIN_04659	None	2.796	up
PIIN_07107	plasma membrane	3.8174	up
PIIN_05757	None	2.1593	up
PIIN_08262	lyase activity	1.7341	up
PIIN_08836	pathogenesis	2.4633	up
PIIN_08345	None	2.9089	up
PIIN_06899	single-organism process	2.3964	up
PIIN_08131	lyase activity	1.8848	up

**Table 4 jof-09-00024-t004:** Functions of target mRNA corresponding to upregulated miRNA in 3 dai vs. mock.

miRNA	Target mRNA	Score	The Function of mRNA
hvu-miR5049a	HORVU1Hr1G020490	0	uncharacterized protein LOC123407470
	HORVU1Hr1G091240	0	Serine/threonine-protein kinase STY13
	HORVU1Hr1G041440	1	uncharacterized protein LOC123431160
	HORVU2Hr1G084120	0	uncharacterized membrane protein YuiD
	HORVU2Hr1G078270	0	protein LURP-one-related 6
	HORVU2Hr1G096290	0	thioredoxin-like 3-3 isoform X1
	HORVU5Hr1G060310	0	putative transcription factor RL9
	HORVU6Hr1G018530	0	uncharacterized protein LOC123402714
	HORVU6Hr1G025780	0	protein argonaute 1B
	HORVU5Hr1G112710	0	leucine rich repeat family expressed
	HORVU7Hr1G096360	0	uncharacterized protein LOC123411193
	HORVU6Hr1G083120	0.5	60S ribosomal protein L35a-1
novel_22	HORVU3Hr1G094730	1	squamosa promoter-binding-like protein 2
	HORVU6Hr1G028980	1	cinnamoyI-CoA reductase 1
	HORVU6Hr1G031450	1	squamosa promoter-binding-like protein 4
	HORVU0Hr1G039170	1	squamosa promoter-binding-like protein 16
hvu-miR171-3p	HORVU4Hr1G010490	0	protein MIZU-KUSSEI 1
	HORVU4Hr1G087700	0	scarecrow-like protein 6
	HORVU6Hr1G063650	1	scarecrow-like protein 27
	HORVU7Hr1G001300	1	scarecrow-like protein 22

**Table 5 jof-09-00024-t005:** Functions of target mRNA corresponding to down-regulated miRNA in 3 dai vs. mock.

miRNA	Target mRNA	Score	The Function of Target mRNA
hvu-miR397a	HORVU6Hr1G025830	0	Uncharacterized protein LOC123403149
hvu-miR444b	HORVU6Hr1G073040	1	MADS-box transcription factor 57
hvu-miR1120	HORVU1Hr1G080480	1	6-phosphogluconate dehydrogenase
	HORVU3Hr1G039220	0	pyruvate kinase, cytosolic. Isozyme
	HORVU3Hr1G067470	1	26S proteasome regulatory subunit 6B homolog
	HORVU4Hr1G002170	1	general negative regulator of transcription subunit 3 isoform X4
	HORVU7Hr1G030930	1	AUGMIN subunit 3
novel_1	HORVU2Hr1G101770	1	growth-regulating factor 3-like
	HORVU6Hr1G068370	1	growth-regulating factor 4-like isoform X1
	HORVU7Hr1G034610	1	growth-regulating factor 2-like
	HORVU6Hr1G081210	1	growth-regulating factor 1-like
	HORVU7Hr1G008680	1	growth-regulating factor 5-like isoform X1
	HORVU0Hr1G016610	1	growth-regulating factor 4-like isoform X2
	HORVU0Hr1G026650	1	predicted protein
	HORVU0Hr1G016590	1	predicted protein

**Table 6 jof-09-00024-t006:** Several key miRNAs targets involved in the regulation of gene transcription.

	miRNA	Gene_names	GO_accession	pval	Description	log2FoldChange
3 dai vs.mock	hvu-miR6189	HORVU3Hr1G055830	GO:0008134	3.0700 × 10^−7^	transcription factor binding	0.94830
	hvu-miR6189	HORVU5Hr1G021690	GO:0001071	3.0700 × 10^−7^	nucleic acid binding transcription factor activity	0.94830
	hvu-miR6189	HORVU3Hr1G086270	GO:0001071	3.0700 × 10^−7^	nucleic acid binding transcription factor activity	0.94830
	hvu-miR6189	HORVU1Hr1G095410	GO:0098531	3.0700 × 10^−7^	transcription factor activity, direct ligand regulated sequence-specific DNA binding	0.94830
	hvu-miR6189	HORVU3Hr1G089580	GO:0000989	3.0700 × 10^−7^	transcription factor activity, transcription factor binding	0.94830
	hvu-miR6189	HORVU3Hr1G026990	GO:0001071	3.0700 × 10^−7^	nucleic acid binding transcription factor activity	0.94830
	hvu-miR6214	HORVU6Hr1G057060	GO:0003700	1.3136 × 10^−2^	transcription factor activity, sequence-specific DNA binding	−1.41320
	hvu-miR6214	HORVU1Hr1G051970	GO:0001071	1.3136 × 10^−2^	nucleic acid binding transcription factor activity	−1.41320
	hvu-miR6214	HORVU7Hr1G036130	GO:0003700	1.3136 × 10^−2^	transcription factor activity, sequence-specific DNA binding	−1.41320
	hvu-miR6214	HORVU1Hr1G020620	GO:0001071	1.3136 × 10^−2^	nucleic acid binding transcription factor activity	−1.41320
	hvu-miR6214	HORVU6Hr1G081340	GO:0003700	1.3136 × 10^−2^	transcription factor activity, sequence-specific DNA binding	−1.41320
	hvu-miR444b	HORVU0Hr1G032300	GO:0001071	5.2200 × 10^−6^	nucleic acid binding transcription factor activity	−2.59460
	hvu-miR444b	HORVU2Hr1G079610	GO:0001071	5.2200 × 10^−6^	nucleic acid binding transcription factor activity	−2.59460
	hvu-miR444b	HORVU2Hr1G080490	GO:0001071	5.2200 × 10^−6^	nucleic acid binding transcription factor activity	−2.59460
	hvu-miR444b	HORVU3Hr1G055960	GO:0001071	5.2200 × 10^−6^	nucleic acid binding transcription factor activity	−2.59460
	hvu-miR444b	HORVU1Hr1G051370	GO:0001076	5.2200 × 10^−6^	transcription factor activity, RNA polymerase II transcription factor binding	−2.59460
	hvu-miR444b	HORVU0Hr1G030830	GO:0003700	5.2200 × 10^−6^	transcription factor activity, sequence-specific DNA binding	−2.59460
	hvu-miR444b	HORVU4Hr1G069340	GO:0003700	5.2200 × 10^−6^	transcription factor activity, sequence-specific DNA binding	−2.59460
	hvu-miR444b	HORVU5Hr1G055470	GO:0001071	5.2200 × 10^−6^	nucleic acid binding transcription factor activity	−2.59460
	hvu-miR444b	HORVU5Hr1G092310	GO:0001071	5.2200 × 10^−6^	nucleic acid binding transcription factor activity	−2.59460
	hvu-miR444b	HORVU5Hr1G119220	GO:0008134	5.2200 × 10^−6^	transcription factor binding	−2.59460
	hvu-miR444b	HORVU4Hr1G087360	GO:0000989	5.2200 × 10^−6^	transcription factor activity, transcription factor binding	−2.59460
	hvu-miR444b	HORVU2Hr1G108210	GO:0000988	5.2200 × 10^−6^	transcription factor activity, protein binding	−2.59460
	hvu-miR444b	HORVU4Hr1G028610	GO:0044798	5.2200 × 10^−6^	nuclear transcription factor complex	−2.59460
	hvu-miR444b	HORVU5Hr1G000370	GO:0001071	5.2200 × 10^−6^	nucleic acid binding transcription factor activity	−2.59460
	hvu-miR444b	HORVU3Hr1G054770	GO:0008134	5.2200 × 10^−6^	transcription factor binding	−2.59460
	hvu-miR444b	HORVU6Hr1G062340	GO:0090575	5.2200 × 10^−6^	RNA polymerase II transcription factor complex	−2.59460
	hvu-miR444b	HORVU6Hr1G066140	GO:0003700	5.2200 × 10^−6^	transcription factor activity, sequence-specific DNA binding	−2.59460
	hvu-miR444b	HORVU5Hr1G123770	GO:0003700	5.2200 × 10^−6^	transcription factor activity, sequence-specific DNA binding	−2.59460
	hvu-miR444b	HORVU5Hr1G000370	GO:0003700	5.2200 × 10^−6^	transcription factor activity, sequence-specific DNA binding	−2.59460
	hvu-miR444b	HORVU7Hr1G024000	GO:0003700	5.2200 × 10^−6^	transcription factor activity, sequence-specific DNA binding	−2.59460
	hvu-miR444b	HORVU7Hr1G023940	GO:0001071	5.2200 × 10^−6^	nucleic acid binding transcription factor activity	−2.59460
	hvu-miR444b	HORVU6Hr1G073040	GO:0001071	5.2200 × 10^−6^	nucleic acid binding transcription factor activity	−2.59460
	hvu-miR444b	HORVU7Hr1G026940	GO:0003700	5.2200 × 10^−6^	transcription factor activity, sequence-specific DNA binding	−2.59460
	hvu-miR444b	HORVU4Hr1G087570	GO:0008134	5.2200 × 10^−6^	transcription factor binding	−2.59460
	hvu-miR444b	HORVU6Hr1G034130	GO:0000989	5.2200 × 10^−6^	transcription factor activity, transcription factor binding	−2.59460
	hvu-miR6190	HORVU1Hr1G082910	GO:0003700	4.6930 × 10^−2^	transcription factor activity, sequence-specific DNA binding	4.59780
	hvu-miR6190	HORVU1Hr1G063610	GO:0003700	4.6930 × 10^−2^	transcription factor activity, sequence-specific DNA binding	4.59780
	hvu-miR6190	HORVU6Hr1G008320	GO:0090575	4.6930 × 10^−2^	RNA polymerase II transcription factor complex	4.59780
	hvu-miR6190	HORVU7Hr1G091040	GO:0003700	4.6930 × 10^−2^	transcription factor activity, sequence-specific DNA binding	4.59780
	hvu-miR6190	HORVU5Hr1G120230	GO:0001071	4.6930 × 10^−2^	nucleic acid binding transcription factor activity	4.59780
	hvu-miR6214	HORVU0Hr1G012230	GO:0000988	4.0994 × 10^−3^	transcription factor activity, protein binding	−3.97730
	hvu-miR6214	HORVU1Hr1G047110	GO:0000989	4.0994 × 10^−3^	transcription factor activity, transcription factor binding	−3.97730
	hvu-miR6214	HORVU7Hr1G001070	GO:0000989	4.0994 × 10^−3^	transcription factor activity, transcription factor binding	−3.97730
	hvu-miR6214	HORVU3Hr1G024950	GO:0003700	4.0994 × 10^−3^	transcription factor activity, sequence-specific DNA binding	−3.97730
	hvu-miR6184	HORVU7Hr1G114030	GO:0001071	3.6984 × 10^−2^	nucleic acid binding transcription factor activity	−4.76840
	hvu-miR397a	HORVU2Hr1G059320	GO:0000989	1.8131 × 10^−2^	transcription factor activity, transcription factor binding	−5.24440
	hvu-miR397a	HORVU4Hr1G080350	GO:0003700	1.8131 × 10^−2^	transcription factor activity, sequence-specific DNA binding	−5.24440
	hvu-miR6214	HORVU1Hr1G020620	GO:0001071	3.0948 × 10^−2^	nucleic acid binding transcription factor activity	0.78887
	hvu-miR6214	HORVU7Hr1G117010	GO:0003700	3.0948 × 10^−2^	transcription factor activity, sequence-specific DNA binding	0.78887
	hvu-miR6180	HORVU2Hr1G035310	GO:0000989	4.8474 × 10^−2^	transcription factor activity, transcription factor binding	−1.65400
	hvu-miR6180	HORVU5Hr1G046390	GO:0001071	4.8474 × 10^−2^	nucleic acid binding transcription factor activity	−1.65400
	hvu-miR6180	HORVU5Hr1G070260	GO:0003700	4.8474 × 10^−2^	transcription factor activity, sequence-specific DNA binding	−1.64310

**Table 7 jof-09-00024-t007:** Examples of miRNAs and their up-and downregulated targets in barley.

	miRNA	Target_ Gene	*p*-Value	GO Term Accession	Function Description	log2 Fold Change
7 dai vs. 3 dai	hvu-miR6214	HORVU7Hr1G036130	3.0948 × 10^−2^	GO:0005179	hormone activity	0.78887
	hvu-miR6180	HORVU1Hr1G037250	4.8474 × 10^−2^	GO:0005179	hormone activity	−1.65400
	hvu-miR6180	HORVU3Hr1G068970	4.8474 × 10^−2^	GO:0005184	neuropeptide hormone activity	−1.65400
	hvu-miR6189	HORVU5Hr1G067480	3.0700 × 10^−7^	GO:0051301	cell division	0.94830
	hvu-miR6189	HORVU4Hr1G036120	3.0700 × 10^−7^	GO:0009725	response to hormone	0.94830
	hvu-miR6189	HORVU3Hr1G089580	3.0700 × 10^−7^	GO:0035257	nuclear hormone receptor binding	0.94830
	hvu-miR6189	HORVU1Hr1G095410	3.0700 × 10^−7^	GO:0015979	photosynthesis	0.94830
	hvu-miR6189	HORVU6Hr1G056490	3.0700 × 10^−7^	GO:0015979	photosynthesis	0.94830
	hvu-miR6214	HORVU2Hr1G021110	1.3136 × 10^−2^	GO:0051301	cell division	−1.41320
	hvu-miR6214	HORVU5Hr1G051090	1.3136 × 10^−2^	GO:0051301	cell division	−1.41320
	hvu-miR6214	HORVU4Hr1G009990	1.3136 × 10^−2^	GO:0015979	photosynthesis	−1.41320
3 dai vs. mock	hvu-miR6214	HORVU5Hr1G070630	1.3136 × 10^−2^	GO:0019684	photosynthesis, light reaction	−1.41320
	hvu-miR6214	HORVU6Hr1G074220	1.3136 × 10^−2^	GO:0015979	photosynthesis	−1.41320
	hvu-miR444b	HORVU5Hr1G092310	5.2200 × 10^−6^	GO:0051301	cell division	−2.59460
	hvu-miR444b	HORVU6Hr1G009230	5.2200 × 10^−6^	GO:0051301	cell division	−2.59460
	hvu-miR444b	HORVU5Hr1G069040	5.2200 × 10^−6^	GO:0047746	chlorophyllase activity	−2.59460
	hvu-miR444b	ENSRNA049488557	5.2200 × 10^−6^	GO:0007059	chromosome segregation	−2.59460
	hvu-miR444b	HORVU2Hr1G124270	5.2200 × 10^−6^	GO:0098813	nuclear chromosome segregation	−2.59460
	hvu-miR444b	HORVU1Hr1G047730	5.2200 × 10^−6^	GO:0005185	neurohypophyseal hormone activity	−2.59460
	hvu-miR444b	HORVU3Hr1G061700	5.2200 × 10^−6^	GO:0005184	neuropeptide hormone activity	−2.59460
	hvu-miR444b	HORVU7Hr1G048310	5.2200 × 10^−6^	GO:0005179	hormone activity	−2.59460
	hvu-miR444b	HORVU2Hr1G090100	5.2200 × 10^−6^	GO:0015979	photosynthesis	−2.59460
	hvu-miR444b	HORVU4Hr1G087360	5.2200 × 10^−6^	GO:0015979	photosynthesis	−2.59460
	hvu-miR6190	HORVU7Hr1G040960	4.6930 × 10^−2^	GO:0005179	hormone activity	4.59780
	hvu-miR6190	HORVU7Hr1G068230	4.6930 × 10^−2^	GO:0005179	hormone activity	4.59780
	hvu-miR6190	HORVU2Hr1G057700	4.6930 × 10^−2^	GO:0042548	regulation of photosynthesis, light reaction	4.59780
	hvu-miR6190	HORVU3Hr1G016330	4.6930 × 10^−2^	GO:0015979	photosynthesis	4.59780
7 dai vs. mock	hvu-miR6214	HORVU3Hr1G021610	4.0994 × 10^−3^	GO:0051301	cell division	−3.97730
	hvu-miR6214	HORVU5Hr1G051090	4.0994 × 10^−3^	GO:0051301	cell division	−3.97730
	hvu-miR6214	HORVU3Hr1G078090	4.0994 × 10^−3^	GO:0005179	hormone activity	−3.97730
	hvu-miR6214	HORVU5Hr1G070630	4.0994 × 10^−3^	GO:0005179	hormone activity	−3.97730
	hvu-miR6214	HORVU7Hr1G036130	4.0994 × 10^−3^	GO:0005179	hormone activity	−3.97730
	hvu-miR397a	HORVU4Hr1G036120	1.8131 × 10^−2^	GO:0009733	response to auxin	−5.24440

## Data Availability

All data generated or analyzed during this study are available in this article and its Appendix A. All miRNA-sequencing data related to the present study have been deposited in the National Centre for Biotechnology Information (NCBI) under the Bioproject accession number PRJNA898289. (https://www.ncbi.nlm.nih.gov/sra/?term=PRJNA898289 (accessed on 12 November 2022)).

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
