# Peer review of "Reprogramming of Fundamental miRNA and Gene Expression during the Barley-Piriformospora indica Interaction"

_jof, 2022, doi:10.3390/jof9010024_

Round 1
Reviewer 1 Report
jof-2060433
Reprogramming of fundamental miRNA and gene expression 1 during the Barley-P. indica interaction
The work behind this manuscript was very well planned, executed and described.
This is an excellent work that deserves to be published on JoF. A compliment to the authors.
However, it requires small corrections and modifications that I suggest below:
Title: the name of the genus is abbreviated too soon P., better to write it in full
L 38 metabonomic metabolomic
L 42 monocot and eudicot better monocotyledons and dicotyledons
L 78 orchid not in italics but in normal font. See also L. 498, 522 (Oncidium italicize the O), 653
L 97 CM it is probably better known as MS instead of CM
L 98 Piriformospora better P.
L 99-103 the inoculation system, while functional, seems rather haphazard without the use of a harmless glue substance. However results have been obtained so it should be accepted.
L 105 (160 µmol m-2s-1, 22 °C) it is not clear what these values refer to
L.115 ‘roots inoculated’ in reality the seeds have been infected don't inoculated the roots, they have been infected
L.116 the word 'dai' appears for the first time and will be repeated many times later. Please explain to the readers in parentheses what this word means. For me it means DNA-dependent activator of interferon-regulatory factors, for you?
L 446 Arabidopsis italics also L515
L 474 “marcomolecule” macromolecules
L 475 activityassociated better activity-associated
Author Response
Dear reviewers ,
We appreciate a lot for the very meaningful and nice suggestions. In response to suggestions andopinions of reviewers, we carefully considered each one and actively improved them. Point-by-point response to reviewers and editors were listed as below
1. Title: the name of the genus is abbreviated too soon , better to write it in full
We used the full name already!
- L 38 metabonomic metabolomic
We corrected already.
- L 42 monocot and eudicot better monocotyledons and dicotyledons
We used monocotyledons and dicotyledons already.
- L 78 orchidnot in italics but in normal font. See also L. 498, 522 (Oncidium italicize the O), 653
We corrected already
- L 97 CM it is probably better known as MS instead of CM.
Here we used CM (complete medium) to culture the P. indica not the MS.
- L 98 Piriformosporabetter
We corrected already
- L 99-103 the inoculation system, while functional, seems rather haphazard without the use of a harmless glue substance. However results have been obtained so it should be accepted.
Thanks for the suggestion. Actually, it is the standard protocol to deal with the seeds and fungi spores.
- L 105 (160 µmol m-2s-1, 22 °C) it is not clear what these values refer to
These parameters refer to the lighting conditions.
- 115 ‘roots inoculated’ in reality the seeds have been infected don't inoculated the roots, they have been infected
Changed to infected。
- 116 the word 'dai' appears for the first time and will be repeated many times later. Please explain to the readers in parentheses what this word means. For me it means DNA-dependent activator of interferon-regulatory factors, for you?
“dai” here means “days after inoculation”.
- L 446 Arabidopsis italics also L515
We corrected already
- L 474 “marcomolecule” macromolecules
We corrected already
- L 475 activityassociated better activity-associated
It is typing problem lack of a blank. Now we corrected already.
Reviewer 2 Report
Title : Please to Indicate in full "-P. indica "
To check and uniform through the manuscript the following acronyms: MiRNAs ( see Lines 64 and 76). Lowercase or capital letter?
Line 23: GO ? what means?
Line 99 Please specify better which seeds of barley line has been utilized
line 100: It is unclear the sentence " seeds...were surface sterilized for 20 min with 3% active chlorine, sodium hypochlorite solution,". Were sterilized twice?
Line 190: change "which indicated that " with "indicating the..."
line 474:marcomolecule?
Please add the paper https://doi.org/10.1073/pnas.0605697103 in the manuscript submitted; some important considerations and information could be interesting for the part of the root colonization.
Author Response
Dear reviewers ,
We appreciate a lot for the very meaningful and nice suggestions. In response to suggestions andopinions of reviewers, we carefully considered each one and actively improved them. Point-by-point response to reviewers and editors were listed as below
1.Title : Please to Indicate in full "-P. indica "
Answer: We used the full name already!
2.To check and uniform through the manuscript the following acronyms: MiRNAs ( see Lines 64 and 76). Lowercase or capital letter?
Answer: We checked already, when it is used in the beginning of a sentence, capital letter was used, the other place, lowercase was used.
- Line 23: GO ? what means?
Answer: “Gene ontology” was added to give the full name of “GO”
- Line 99 Please specify better which seeds of barley line has been utilized
Answer: “JINNONG 8” was used in our study and added in the MS.
- line 100: It is unclear the sentence " seeds...were surface sterilized for 20 min with 3% active chlorine, sodium hypochlorite solution,". Were sterilized twice?
Answer: This sentence means to sterilize the seed surface with sodium hypochlorite containing 3% active chlorine for 20 minutes
- Line 190: change "which indicated that " with "indicating the..."
Answer: Change already.
- line 474: marcomolecule?
Answer: We corrected it to “macromolecules”
- Please add the paper https://doi.org/10.1073/pnas.0605697103 in the manuscript submitted; some important considerations and information could be interesting for the part of the root colonization.
Answer: This literature was added according to the reviewer’s suggestion.